# Learning and Developing Together for Improving the Quality of Care in a Nursing Home, Is Appreciative Inquiry the Key?

**DOI:** 10.3390/healthcare11131840

**Published:** 2023-06-24

**Authors:** Regula Van Graas, Robbert J. Gobbens

**Affiliations:** 1Faculty of Health, Sports, and Well-Being, Inholland University of Applied Sciences, De Boelelaan 1109, 1081 HV Amsterdam, The Netherlands; robbert.gobbens@inholland.nl; 2Zonnehuisgroep Amstelland, 1186 AA Amstelveen, The Netherlands; 3Department Family Medicine and Population Health, Faculty of Medicine and Health Sciences, University of Antwerp, 2610 Antwerp, Belgium; 4Department of Tranzo Academic Centre for Transformation in Care and Welfare, Faculty of Behavioural and Social Sciences, Tilburg University, 5037 AB Tilburg, The Netherlands

**Keywords:** appreciative inquiry, quality of care, nursing homes, learning, developing

## Abstract

To investigate the impact of Appreciative Inquiry (AI) on learning and developing together for improving the quality of care in a nursing home, and to explore experiences following the four phases of the AI cycle, an AI was performed as part of participative action research. Interviews, group discussions, creative methods of storytelling, and reflections were used in the AI sessions. Staff members were interviewed to evaluate the AI process. All of the interviews were recorded and transcribed. Data were analyzed thematically using Maxqda and were discussed by the interviewers until consensus was reached. The sessions resulted in ten action proposals to improve healthcare. One proposal was worked out in agreements on cooperation. The evaluation revealed that the sessions have given a boost to the team spirit, and involvement at the location leading to more cooperation and communication during the shifts. The evaluation indicated that it is important to convert the action plans of the sessions into actions, which are then evaluated. AI was advised as a way of learning where everyone is involved, and mutual agreements are made. This study has shown that AI can be a valuable way to support learning and development to promote the quality of care in a nursing home.

## 1. Introduction

Care in nursing homes is becoming ever more complex, due to the frailty and multimorbidity that many of these residents suffer from. This also makes care in nursing homes time-consuming [1]. Keeping continuity in healthcare is a major challenge that affects service quality and safety. Rolland et al. [2] report a high number of vacancies and a high turnover rate of nurses and nursing assistants due to underpayment, lack of recognition, lack of resources, and frequent confrontation with vulnerable families who are under pressure from psychological and economic conditions.

The provision of person-centered care, which is a dominant indicator of high-quality care, is not easy to achieve [3]. Rutten et al. [4] mention the following necessities to ensure good quality of care: transformational leadership, four job characteristics (i.e., social support from a leader, work satisfaction, task variation, and development opportunities), unity in the philosophy of care, and strengthening teamwork by facilitating relationships and collaboration between care staff, leaders, the residents, and their family. 

There is a growing international awareness of the need to perform high-quality care in nursing homes [2]. Skilled geriatric competences are required from the whole staff to fulfill all of the needs of residents in a nursing home: not only to fulfill their specific medical care needs but also their psychological and social needs, and to improve their quality of life by providing a pleasant, homelike environment, meaningful activities, and social contacts [2]. The question of how to maintain or even improve the quality of care in nursing homes remains topical. Handley et al. [5] showed that learning general theory about dementia is not good enough, knowledge should be relevant, practical, and applicable [6]. Involving staff in the development of practice may serve as a useful strategy for producing relevant knowledge [7]. Van Lierop et al. [8] call on these organizations to facilitate workplace learning and for long-term care organizations to become instead learning organizations. Workplace learning is defined as informal learning, sometimes combined with formal learning, which takes place during daily practice [9]. Research in learning organizations distinguishes multi-levels in the organization to be important. Gavin et al. [10] revealed the three building blocks of organizational learning; leadership that reinforces learning (engage in active questioning and listening, demonstrate a willingness to entertain alternative viewpoints), learning processes (experimentation, information collection, and transfer, education, and training, analysis) and a supportive learning environment (psychological safety, openness to new ideas, time for reflection, appreciation of differences). Decuyper et al. [11] mentioned the levels of organization (strategy, involved leadership), the team (team leadership, management), and the individual (self-efficacy, being flexible and motivated). 

The above elements of learning can also be found in the self-determination theory of Ryan and Desi [12]; people’s intrinsic motivation can be increased if three basic psychological needs are addressed: Autonomy, sense of competence, and social connectedness. Autonomy is about choices and influence. The sense of competence is the confidence in your own ability. Social connectedness is the connection with the environment, the confidence in others, and a positive and safe climate where you feel free to ask questions and you are not afraid to make mistakes. 

While focusing on long-term care for older people, Van Lierop et al. [8] mapped the conditions of workplace learning at individual, team, and organizational levels, including: facilitating characteristics (e.g., to be given time and room for (team) development), behavioral characteristics (e.g., an open attitude and feeling responsible), context and culture (e.g., feeling safe), cooperation and communication (e.g., giving and receiving feedback), and knowledge and skills (e.g., acquiring knowledge from each other and sharing knowledge). Dewar and MacBride [13] state that for nursing homes, further development of learning interventions is required that focuses on work-based educational models. The focus should be on actively involving healthcare professionals in a way that is comfortable, taps into their strengths and capacity, and helps them to think creatively about possible solutions that mean something to individuals [14]. 

Appreciative Inquiry (AI) is a promising approach that is able to bring people together, and for them to connect openly and honestly with each other to achieve change together. AI is defined as a narrative-based process of positive change [15]. AI is a cycle of four phases (i.e., 4D-cycle) that starts by engaging all members of an organization or community in a broad set of interviews and entering a deep dialogue about strengths, resources, and capabilities (discovery). People are then put through a series of activities that are focused on envisioning bold possibilities and lifting up the most promising dreams for the future (dream). From there, it asks people to discuss and craft propositions that will guide their future together (design). Finally, it forms teams to carry out the work needed to realize new dreams and designs for the future (destiny) [15].

There are five principles underlying a good AI process: positive, constructionist, simultaneity, poetic, and anticipatory [16,17]. First, “positive” refers to positive imagery that has a therapeutic effect. Questions that seek to identify strengths engage people in positive change. “Constructionist” refers to the reality that is created in communications, words, and dialogue with others. The narrative is a stimulus for change. The “simultaneity” principle considers that inquiry and change are not separate processes. Change begins in the first questions asked. The fourth “poetic” principle pays attention to reframing current views of a situation to ensure that there is a basis for creativity and innovation. Finally, “anticipatory “refers to envisioning future potential changes that create a feeling of control over a difficult situation and enhance their motivation to change the situation. AI provides a positive and new way of participating in healthcare and health research, which is often described as engagement, involvement, and inclusion. This presents nurses with an opportunity to develop effective social networks, high levels of engagement, and interdisciplinary collaboration [18], from a bottom–up perspective [19], that supports change in nursing practice [20].

AI is increasingly being used as a method to conduct research into healthcare and as a way to achieve change in healthcare in general [17]. It is also used within long-term care in nursing homes to (for example) explore the education and developmental needs of care home nursing staff [21] or to develop caring conversations in the care home setting [13]. AI has also proven its usefulness as a knowledge translation strategy because it ensures that knowledge from research is implemented in the workplace [17]. Watkins et al. [17] performed an integrative review of the impact of AI on changing clinical nursing practice in in-patient settings. They found only eight studies, between the years 1990 and 2015, that used the complete 4D-cycle—one study had achieved a transformation, while the other studies resulted in small changes in nursing practice and behavior. They concluded that there is only a limited application of AI principles overall, with inconsistencies in the operationalization and reporting, which makes judgments regarding the impact in nursing difficult [17]. For AI to actually succeed (i.e., to bring about a change in healthcare), it is necessary to appropriately apply the four phases and the five principles of AI as stated above [17,19].

AI is an upcoming approach in the field of nursing homes in the Netherlands. For example, Sion et al. [22] advise using the 4D-cycle of AI for nursing staff to help them reflect on and learn from narrative qualitative data from the Connecting Conversations project, together with families and residents on an operational and tactical level. Sion et al. [22] indicate that this use of the 4D-cycle of AI has yet to be investigated.

To the best of our knowledge, there have been no studies using the complete 4D-cycle and the principles of AI in Dutch nursing homes to learn about and develop the quality of care. Therefore, the findings from this study contribute to knowledge regarding AI as a method to learn. The goals of this research are: (a) to investigate the impact of a complete AI cycle on learning and improving the quality of care in a nursing home, and (b) to explore the experiences following the entire cycle of AI with a care team in a Dutch nursing home.

## 2. Materials and Methods

### 2.1. Design

Within a participative action research design (PAR), AI was chosen as the method for this study. AI, such as PAR, is an action research methodology that involves its subjects as co-researchers [23], is based on critical reflective practice [15], and fosters collective action and social change [24]. In distinction to conventional action-research, the knowledge-interest of AI lies not so much in problem-solving but in social innovation from an appreciative approach [25]. The four phases of AI were completed in three sessions, and the five principles of AI were applied appropriately [15]. The sessions also incorporated the elements of AI group sessions as described by Masselink et al. [26]: ‘whole system in the room’ (i.e., everyone who has an interest in the topic participates), striving for a plurality of perspectives, empowerment for all, assuming self-direction, and finally seeking broad agreement for decision-making.

### 2.2. Context and Participants

The study took place from September 2021 until July 2022 in Bovenkerk, a nursing home in the Netherlands that belongs to Zonnehuisgroep Amstelland. This organization was chosen because the second author is affiliated with the organization as a professor of Health and well-being of the frail elderly. Furthermore, the organization encourages conducting scientific research in the care of older people.

This nursing home is formed of eight units, which are registered for 48 residents with dementia and employ 90 staff members. This nursing home uses the concept of small-scale living where eight residents live in one unit in a family atmosphere, with a living room and a private bedroom for each resident. The care is provided by a diverse group of caregivers—mainly healthcare assistants, certified nurse assistants, and a few registered nurses. The care is supported by ward assistants and activities coordinators. Management, other disciplines, such as a geriatrician, psychologist, physiotherapist, and also facility staff are also present at the location.

The project’s researcher introduced the concept of AI to the workgroup of the project as a promising way of learning and improving. This workgroup included the management of the location, the researcher, the team coach, the quality manager, the practical trainer, the activity supervisor, and one team member (a registered nurse). To prepare and facilitate AI, a trio was composed of the workgroup that included the researcher, the team coach, and a team member. In this way, knowledge about AI and research (researcher), group processes (team coach), and information about the team members and connection with the team (team member) were brought together.

All of the employees at the location were invited to participate in three AI (team) sessions. However, because of potential problems with the continuity of care for the residents, not everyone could get out of care at the same time, and therefore two parallel sessions were always organized.

### 2.3. Data Collection

Data were collected throughout the AI process by the researcher, the facilitators of the AI process, the participants in the team sessions, and the workgroup of the research project. A variety of (qualitative) data collection methods has been used as appropriate to the method of AI, including interviews (e.g., discover), group discussions (e.g., discover, design, destiny), creative methods (dream), and storytelling (e.g., discover, dream). Outcomes were written on whiteboards, which have been preserved, and were worked out by the facilitators. Recordings of the explanation of the dream phase were transcribed. The data collection methods of the AI sessions are outlined in Table 1.

To evaluate the AI process, the three facilitators interviewed staff members. To ensure that we got a representative sample from the participants of the AI sessions, some respondents were chosen deliberately, the psychologist, the activity coordinator, and management, for example. The interviews of the care personnel were completed with employees who were working when the facilitators were on site (convenience sample). Then the planning was checked to plan interviews with the care workers who had not yet been interviewed. This was continued until data saturation was reached. Care workers were interviewed individually or, for efficiency reasons, in small groups. See the specification of the interviews in Table 2.

Semi-structured interviews were chosen to enable the participants to ask questions if necessary [27].

An interview protocol was drawn up to ensure the three facilitators conducted the interviews in the same way, which benefits the quality of the data (see Appendix A).

The three facilitators themselves also did an evaluation of the AI sessions by interviewing each other based on the topic list that was also presented to the team members.

All interviews were conducted orally, audiotaped, and then transcribed. This study is conducted following the Standards for Reporting Qualitative Research (SRQR) [28].

### 2.4. Data Analysis

#### 2.4.1. AI Process

The data analysis involved a participative approach based on Appreciative Inquiry strategies, such as reflection, and deductive and inductive coding [29]. In each AI session and in each AI phase, the facilitators reflected orally with the participants on the data using appreciative questions such as: Is this a good representation? Is it complete? Is there anything missing? Does anyone have any reservations? Can everyone agree with this? In this way, the data was immediately member-checked [30] and was therefore validated and served as input for the next AI phase.

In addition, the facilitators and the working group reflected on the outcomes and the process of both parallel sessions going through the data, interpreting possible meanings, and identifying important themes. These themes were brought back into the AI sessions to check whether the interpretation of the facilitators and working group matched the interpretation of the team members.

#### 2.4.2. Evaluation of AI

The steps of thematic analysis by Braun and Clarke [31] were used to analyze the data from the interviews. The researcher openly coded the interviews in Maxqda 2022 and put them in themes. The other two interviewers read the transcripts and reviewed the codes and the themes the researcher made in Maxqda. The codes and themes were discussed by the three interviewers until a consensus was reached.

### 2.5. Rigor

Rigor was considered in different ways in the research. In the setup of the research, a specific trio of facilitators was chosen, namely the researcher, the team coach, and a team member of the location. In the process of data collection and data analysis, they were able to inspire, check, and correct each other. Secondly, an experienced external AI facilitator was coaching the three facilitators, ensuring the careful completion of the analysis steps in an AI phase and the proper follow-up of the various AI phases was guaranteed. A third step was to ask the working group to what extent the outcomes of the AI sessions were recognized. They confirmed that the outcomes were consistent with the experiences on the location.

During the AI sessions, attention was also paid to rigor by asking the participants in all steps whether they recognized themselves in the (intermediate) outcomes (member check) (Birt et al., 2016). Finally, the steps and considerations in data decisions have also been recorded and described for reasons of transparency and replicability of the research. To keep the article readable, those steps and decisions are described in the results.

### 2.6. Ethical Considerations

For this study, medical ethics approval was not necessary because particular treatments or interventions were not offered or withheld from respondents. The integrity of the respondents was not encroached upon as a consequence of participating in this study, which is the main criterion in medical-ethical procedures in the Netherlands [32]. Permission has been granted by the healthcare institution Zonnehuisgroep Amstelland to start the participatory action research at the location. Employees of Bovenkerk were informed about the purpose of the study and given informed consent to participate in the study.

## 3. Results

### 3.1. The Preparation Phase

In the workgroup, the researcher explained the 4D-cycle and the principles of AI. The members experienced AI by interviewing each other appreciatively about working together. A positive theme was then established for the AI sessions using an appreciative exercise. The members first filled in a problem tree (with a problem in the trunk, the causes of the problem in the roots, and the effects of the problem in the crown). The perspective tree was then filled in (with what the location wants to pursue written in the trunk, what contributes to this written in the roots, and the fruits of this theme in the crown). The chosen theme was: “Together, we provide the best care for our residents”.

The team sessions were scheduled, and invitations were sent out to all of the employees of the location. An employee could sign up for one of the two parallel sessions. Since team members could freely register for a session, the number of participants in a session was variable. The original intention was to schedule the three sessions six weeks apart. However, due to the COVID-19 pandemic, the sessions were held in September 2022, and in May and July 2023. Sessions one and two were held outside the nursing home and lasted five hours (including dinner). Session three was organized in the nursing home and lasted two-and-a-half hours.

The facilitators prepared the sessions in which they were coached by an experienced AI facilitator, this was the AI trainer, whereas the researcher followed a basic AI course.

### 3.2. Discovery

In September 2021, two parallel team sessions were planned. The sessions included 25 and 45 participants, respectively. After an explanation about the 4D-AI cycle, we formed groups of two who held an appreciative interview with each other about situations where they had provided good care for the residents and their families and worked well with their coworkers. The pairs were composed of one of the facilitators, the team member, so that as many different perspectives as possible would meet in the interview, taking into account the degree of courage of team members to speak out. Each participant was given an A4 with instructions on it containing the themes to be covered and examples of appreciative questions. Instructions were also given as to what the interviewer should write down from the interview (for these instructions, see Appendix A). Both interviewers wrote down the criteria they heard for good care and working together with family and in the care team. Then the facilitators combined four pairs at random into subgroups of eight people. Each participant told the subgroup the criteria they had heard in the interview. A facilitator guided this process so that everyone was heard, and in-depth questions were asked. One subgroup member wrote down all the criteria. After everyone had their turn, duplicate criteria were identified and removed, sometimes after discussing whether similar terms were indeed the same, duplicate terms were removed. Finally, the subgroup arrived at the top three most important or valuable indicators on each of the three themes; a top three that everyone agreed with. Then the same process took place with the whole group present, where under the guidance of the facilitators, another top three was made with the most important and most relevant indicators per theme for the whole group.

### 3.3. Dream

In the same team session, new groups have been created based on the element of AI group sessions to get as many perspectives as possible (‘striving for a plurality of perspectives’). Each group made an artwork with the following dream question in mind: What is possible for this nursing home if we meet the criteria that we just agreed on in the discovery phase? The subgroups had a variety of creative materials at their disposal. In two sessions, seven artworks were made. Each group told the story of their artwork: What is it about? Why this artwork? The others listened and asked questions. The facilitators encouraged participants to ask questions or asked in-depth questions themselves to the presenting groups to ensure that the meaning of the artwork was well understood. Figure 1 presents some of the artwork and quotes of the stories told.

After the first two sessions, the facilitators reflected on the outcomes of the discovery phase and combined the lists of criteria Duplicates were removed. Indicators that were similar but slightly differently formulated were put back in a team meeting to check whether they were indeed the same terms (see Table 3).

### 3.4. Design

In May 2022, two parallel team sessions were planned. The sessions included 25 and 35 participants, respectively. To build upon the previous AI sessions, we started by showing the videotaped recordings of the seven artworks from the dream phase. We formed random teams of five persons and asked them to write down the themes that connected the artwork. The subgroups presented the themes to each other. The facilitators asked the subgroups to make one single list that everyone could agree on. In total, 10 themes were identified (see Table 4). Due to the parallel sessions, there is an overlap in the themes.

The participants then chose a theme that they wanted to develop into a change proposal. Each group answered 10 appreciative questions to clarify what needs to be achieved on the chosen theme, what is already going well in the theme, and what needs to be worked on (see Table 5 for the questions).

After the two sessions, the facilitators discussed the change proposals in the workgroup of the research project. It was jointly decided to develop the theme of cooperation and communication in the destiny phase because this theme emerged in several change proposals and could serve as a basis for developing the other proposals.

### 3.5. Destiny

The third AI session was held in July 2022. The sessions included 25 and 35 participants, respectively. The participants first individually completed a questionnaire regarding the learning climate of the location. This questionnaire [33] is a free translation of Edmundson’s [34,35] questionnaires on psychological safety and team learning with questions about goals, development, teamwork, safety, and communication (see Appendix A), used in this healthcare institution to start coaching teams, to work better together. The participants discussed the answers to the questionnaire in small groups (n = 4) and wrote on whiteboards what was already going well in the communication and what could be improved. The subgroups then presented their whiteboards to each other. This resulted in honest conversations about how the collaboration was experienced now and how they would like to see it. Subsequently, each subgroup made a top five important collaboration agreements, which, after agreement throughout the group, resulted in a list of collaboration agreements that the entire group supported (see Table 6).

In each session, two participants were asked to turn the two overviews from both sessions into one single list of agreements that would be appropriate and relevant for the entire location. The researcher supervised this process. This eventually resulted in nine collaborative agreements. These nine agreements were presented to the working group and discussed in the team meeting. Everyone on the team agreed with these nine agreements. The location had these agreements printed on posters and pocket cards, see Figure 2.

The colleagues who contributed to the final agreements acted as the owners of this process and put up the posters, handed out the cards to team members, and drew attention to the agreements at meetings and handovers. They also took it upon themselves to evaluate in due time whether the team members kept to the agreements and whether this has improved the atmosphere in the team.

After the three sessions, it was agreed in the working group that the management would take on the responsibility, together with the team, to formulate proposals, based on the outcomes of the AI sessions, leading to actual change in the workplace. The facilitators offered coaching to the management to ensure that this process is completed in an appreciative way.

### 3.6. Evaluation of AI

The three facilitators of the AI interviewed 30 employees of the location.

The interviewers asked about the experience of the sessions, whether the sessions contributed to the goals of the sessions (e.g., better cooperation and atmosphere, knowing each other better, being heard, and motivation to do better in care for the residents), whether necessary actions to continue the process of AI at the location and if they would recommend AI as a method to learn for the location.

#### 3.6.1. Experiences

The employees generally look back positively on enthusiastic, relaxed sessions, especially the first session in which they used their creativity to dream and craft this dream. They were also asked to think about what is desired instead of what is not possible or not going well:


*I found it very fun and fascinating. Different from a normal consultation. You did see people in a different way. It was a bit more playful than a normal meeting and you saw colleagues in a different light than usual... It was nice to think creatively and a bit looser and not in ’what is not possible,’ but what would we really like.*

*(Interview 5, conversation with care worker)*


They enjoyed being with each other. The sessions were often described as instructive because opinions were shared, brainstormed, and discussed together:


*You can immediately let the whole team know what can be improved or not. Then you immediately get a discussion, brainstorming and that’s good. Yes, exchange of ideas, experiences.*

*(Interview 14 with care worker)*


#### 3.6.2. Goals of the Sessions

The participants have gotten to know each other better and felt more connected to each other:


*I have seen in those team sessions that people showed more who they were.*

*(Interview 17 with coordinating activity supervisor)*



*I think especially the piece of connection. Not really something very concrete, but more connecting together ...... the feeling. Equal care teams and practitioners: we do this together.*

*(Interview 18 with psychologist)*


Most team members mentioned openness and honesty in the conversations. In general, team members felt free to express their opinions in the sessions, and felt that they were heard:


*Even if someone did not agree with it, it was still said.*

*(Interview 6 with care worker)*



*I think everyone misses discussing with the others. And those sessions are very good for that. Everyone listened attentively.*

*(Interview 14 with care worker)*


The sessions have given a boost to the team spirit. The atmosphere and involvement have been improved at the location:


*I do think there was a need to change the working atmosphere. I do think that appreciative change has contributed to the team spirit, team building, connection, support for each other’s opinions, and ideas.*

*(Interview 16 with management)*



*If you look at how we were before those sessions, and where we are now as a team, they are two different teams.*

*(Interview 12 with care worker)*


Most healthcare workers indicated that the sessions motivated them to change their behavior (e.g., more cooperation, helping each other, addressing each other, communicating more, and having more understanding for each other):


*It has become more helpful, [there is] more cooperation.*

*(Interview 8 with four care workers)*



*Ask sooner, call sooner. Just do it, because we are here for each other. I try to do that more now.*

*(Interview 1 with care worker)*


Two care employees reported that agreements have now been made with each other and that they can talk to each other about this.

The management takes from the sessions that nothing is impossible, that it is important to question the desire behind a story and to align the way of learning with how the team wants to learn:


*I use the phrase very often: I hear what you say, but what is the wish behind it?*

*(Interview 16 with management)*



*I especially learned how the team wants to learn, especially the how. How do they actually want to learn? Did we offer it the right way?*

*(Interview 16 with management)*


#### 3.6.3. How to Continue the Process of AI

Suggestions were mentioned in the following four themes: do not just talk but act, keep naming and discussing, come back to agreements made, and finally a clear role for management with a clear division of roles between management and any others.

For “not just talk but act”, it is important to follow plan-do-check-act, in which actions are set out and are also evaluated. That is not happening enough now:


*We have now implemented it and then we have to see how things are going in practice. We can get information from that: how are things going now? Is it well implemented? Is it clear to everyone?*

*(Interview 16 with management)*


To “keep naming and discussing” it, it is important to create moments where team members see each other and discuss the actions. It is also mentioned to take small steps:


*That you occasionally have those action points come back: remember, we were working on that and that.*

*(Interview 12 with care worker)*



*If you have eight steps, first try to complete step one. And then the next step.*

*(Interview 6 with care worker)*


It is also important to come back on joint agreements, take joint responsibility for this, and also hold each other to account:


*Because we did it together anyway. So, make it your own and if everyone takes their responsibility to make it their own and then we can give each other better feedback. The agreements were made together.*

*(Interview 16 with management)*


Finally, a clear role is expected from management with a clear division of roles between management and any initiators of certain plans that may be appointed. This is currently insufficiently clear and visible:


*A clear course. People want clarity.*

*(Interview 4 with care worker)*



*But there isn’t really a division of roles yet.*

*(Interview 16 with management)*


#### 3.6.4. Recommendation to Use AI as a Way to Learn

In the interviews with both care workers and management, it is indicated that AI is a way to learn and develop as a team:


*It puts me back on my feet every now and then: oh, these are their agreements. If we had not done this, it would not have been their agreements, but more from what we thought from ‘that ivory tower’ would have been good.*

*(Interview 16 with management)*



*Would I advise appreciative change? Always, the part of team building. Despite the fact that you work one-on-one, you hear other people. At least, I hear what other people think.*

*(Interview 6 with care worker)*



*I would advise it. You also have all disciplines and colleagues. you think together, make agreements together. ….. And that everyone gets the opportunity to be involved.*

*(Interview 16 with management)*


#### 3.6.5. Evaluation Facilitation

The facilitators also evaluated the AI process themselves using the same interview questions that were also asked of the employees, which resulted in four main points. The first point is the enthusiasm about how the AI brought energy to the team sessions and what AI has subsequently yielded for the location. The second point is the success of facilitating in a trio where knowledge of the method (researcher), group processes (team coach), and the team (team member) came together. The trio also made it possible to discuss during and between sessions to see if the process was going well. The third point is not only the importance of proper preparation and facilitation of the sessions but also the importance of executing the different phases of AI in the right way. It was a good choice to do the sessions outside the nursing home because the choice of location turned out to have had quite an influence on the course of the sessions (e.g., the size of the room, the acoustics, the possibility to hang whiteboards on the walls and the ability to move furniture). Being able to walk in the room, standing by whiteboards, and having space to work in subgroups has been a success factor. Finally, the fourth point is that the action plans were not implemented on their own, the facilitators looked for a more steering role from the management, but this was not sufficiently the case so far.

## 4. Discussion

The participants looked back on enthusiastic AI team sessions that resulted in good conversations, a better team atmosphere, and connectedness by thinking together about what good care should look like and how to get there together. These experiences are similar to the results of the review study on AI by Watkins et al. [17] describing trust, dialogue, teamwork, and the uplifting effect of focusing on strengths while going through an AI cycle. A good atmosphere and a safe learning climate are prerequisites for learning. For example, Garvin et al. [10] describe a supportive learning environment as one of the three building blocks of a learning organization, which includes psychological safety, appreciation of differences, openness to new ideas, and time for reflection. The AI sessions served as a time for reflection that “allow time for a pause in the action and encourage thoughtful review of the organization’s processes” [10] (p. 3). The conversations with each other have led to changes in cooperation during the provision of care, which the principle of simultaneity already promised [17]. They learned from and with each other by reflecting on how they currently work together in healthcare and by exchanging experiences and ideas (e.g., about person-centered care). This is characterized by Watkins et al. [17] as a knowledge translation strategy.

The client or their representatives were not present during the sessions, which is advised according to the principles of AI [15]. This was deliberately deviated from because it did not feel safe for the team to question their cooperation in the presence of the residents’ families. Therefore, it is recommended to bring care and resident or family into dialogue with each other in the follow-up process of AI on topics that concern the resident or family.

Although useful action proposals have been made, concrete actions have not been forthcoming so far. Looking back, the destiny phase was insufficiently elaborated because it should have been made clear who should pick up what and when. This has led to delays and to the feeling of the team that nothing is happening regarding their action proposals. It is important here not to make a master plan that cannot be deviated from because the strength of AI is precisely that it moves along with what is considered urgent by the participants: “Therein is a caution against the choreography of Appreciative Inquiry where participant experiences or stories are molded to fit an agenda or a previously drafted master plan” [17] (p. 2). However, it remains up to management to facilitate and monitor to achieve learning, as described by Garvin et al. [10] in their third building block of a learning organization. In particular, Dewar and Nolan [36] stated that management support is of great importance for the implementation of AI.

What has worked powerfully is facilitating with three people: a team member, the team coach, and the researcher; bringing together knowledge of the method (researcher), team processes (team coach), and the team (team member). This gave us the opportunity to discuss the process, and the right substantive steps and/or interventions in the preparation for, and guidance in, the sessions. The team member and team coach can give the team a sense of security because they know them. It also offers the possibility of a smooth transition from the process to the organization or location. 

Good facilitators are of great importance for the process and outcomes of an AI [17,19]. The coaching sessions that the researcher had from an experienced AI facilitator were of considerable value and influenced the outcome of the sessions positively. However, the three facilitators were not experienced facilitators, so they may not have been able to get the most out of the sessions. In addition, due to the COVID-19 pandemic, there was a lot of time between sessions, which meant that we had to spend a lot of time retrieving results from previous sessions. This repetition was at the expense of depth in the sessions.

The AI process has been completed in close consultation with the working group and the participants, which guarantees the participatory aspect of the research [37]. However, the triangulation of qualitative data collection methods may have provided the data with a deeper meaning and helped to minimize the limitations of each method [27]. The reliability of the findings was increased by a continuous member check in the sessions, in which the outcomes of a session were always coordinated with those present and thus validated [30]. Due to the methods used in the AI sessions, the facilitators have strived as much as possible to hear all opinions, to understand what is being said by asking many questions, and, finally, to arrive at supported decisions. To check whether everyone’s opinion has really been taken into account, analyzing all completed forms and whiteboards would have been another option. We, therefore, recommend this check for a subsequent study where AI is used.

The AI process was followed at one location with one team, which makes the generalizability of the workable principles of AI small. Therefore, it would be valuable to do follow-up research in more locations and teams to see if the AI process leads to the same experiences and outcomes.

Due to the termination of the research project after the third AI session, we were, unfortunately, unable to retrieve data from the evaluation that would have been completed after the implementation of the agreements made about communication and collaboration. With this evaluation, we could have said more about the impact of going through the AI process.

Watkins et al. [17] propose using the appraisal tool by Bushe and Kassam [38] to measure the impact of an AI process, in particular: Have assumptions changed? Has new knowledge been generated that has caused a radical shift in the way things are done? And, has a platform been found to move forward, for example by entering into and resolving conflicts? Judging by this appraisal tool, the impact of going through the AI cycle in this study is small because it is not yet possible to make firm statements about the improvement of care. However, there are proposals for change that can form the basis for further professionalization of the team and the improvement of care. It is up to the location to continue to use the AI philosophy and to work together and learn in this way for the benefit of the quality of care.

## 5. Conclusions

To provide good care to residents in a nursing home, it is necessary to continue learning—both as an individual and as a team. AI is increasingly being used as a method to effect change in healthcare. This study investigated the experiences of a nursing home’s teams while going through the entire AI cycle and asked whether this has resulted in an improvement in care. The participants experienced the AI method as an enthusing method that brought colleagues together and created connections by looking together at where people want to go and how to get there. Going through the AI cycle has resulted in a better atmosphere in the team with increased psychological safety, which is a prerequisite for learning [10]. This study has shown that AI could be a valuable way to support team learning and development. Follow-up research in Dutch nursing homes on the use of AI as a learning and development method is recommended to collect more evidence about the process and impact of AI.

## Figures and Tables

**Figure 1 healthcare-11-01840-f001:**
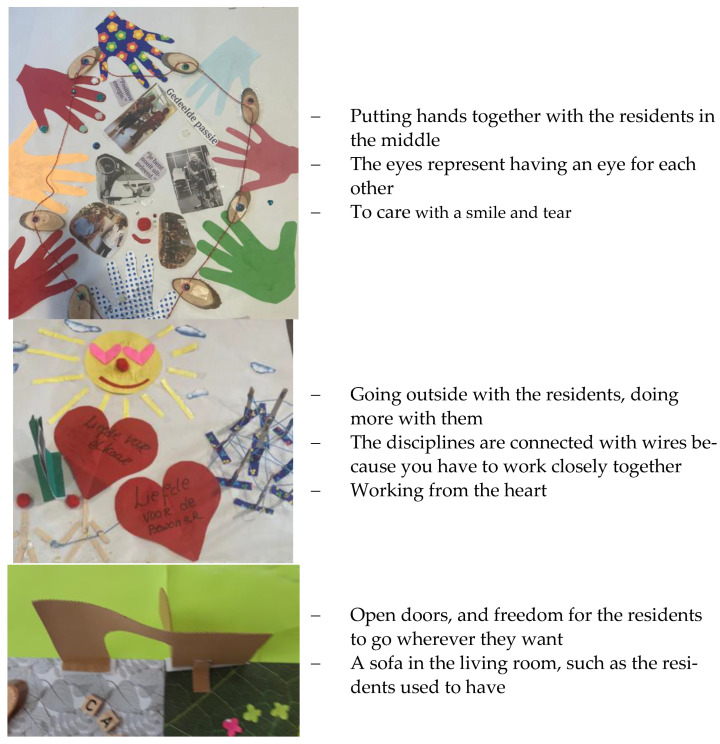
Examples of artworks from the dream phase.

**Figure 2 healthcare-11-01840-f002:**
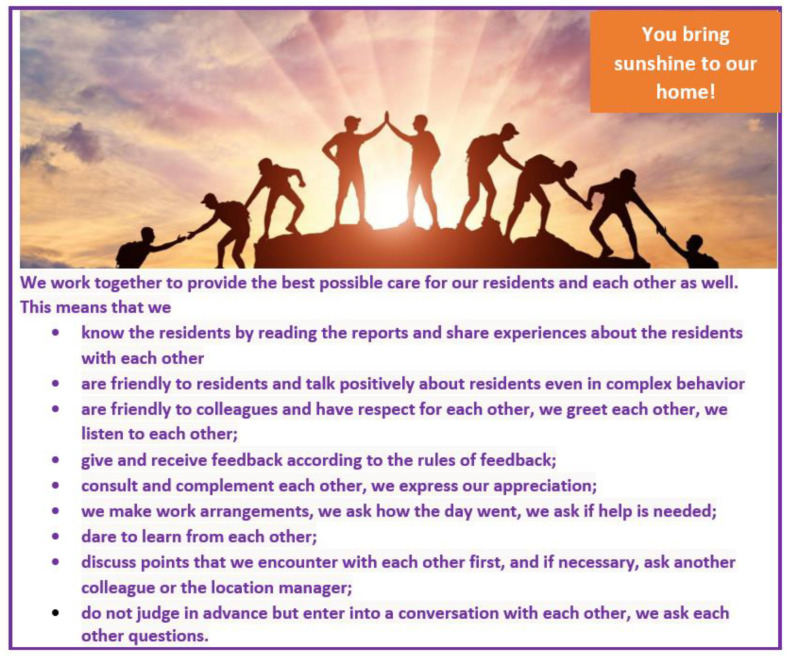
Poster with agreements on working together.

**Table 1 healthcare-11-01840-t001:** Overview of AI phases and data collection.

Phase AI	Data Collection
Discover	Individual interview notes from the interviews in pairs (25 in the first session and 45 in the second session) Seven summaries of key indicators from the interviews were made in pairs by subgroups of eight people (three in the first session and four in the second session) Per session one paper with the key indicators for good care and good cooperation with the family and as care team, that applies to the attendees One summary paper with the indicators of good care and good cooperation with family and the care team from both sessions
Dream	Seven artworks from two sessions (three in the first session and four in the second session) Seven stories about the artworks captured as images and transcripts
Design	Eight elaborations on paper of themes that those present have taken from the film shown, containing the seven works of art from the dream phase (four in the first session and four in the second session) Two summary papers with themes taken from the film by the attendees in a session about the seven works of art of the dream phase (one per session) Ten change proposals (five from each session)
Destiny	Six notes from subgroups on what is going well in the collaboration and communication and what can be improved (three from each session) Six elaborations of a top three with cooperation agreements (three from each session) Two summary papers with cooperation agreements by the attendees in a session (one per session) One list of cooperation agreements as a result of both sessions

**Table 2 healthcare-11-01840-t002:** Specification of interviews to evaluate the AI sessions.

Interviewer	Interviewees
Team coach	Interview with one care member Three interviews with two care members Interview with a care member and a facility employee Interview with four care members Interview with a ward assistant
Team member	Interview with the psychologist Interview with the activity coordinator
Researcher	Six interviews with one care member Interview with a care member and a facility employee
Researcher, team coach	Interview with management: location manager, two team leads, and a quality-nurse
Researcher, team coach, team member	Interview with each other in which two members have interviewed the third member

**Table 3 healthcare-11-01840-t003:** Combined list of criteria in the themes of good care, working with family, and working together.

Good Care for the Residents	Working with Family	Working Together in the Team
To know the residents Respect the wishes of the residents Take time for the residents A peaceful atmosphere Working from the heart	To respect and to trust each other Get to know the resident by talking to the family To express and discuss expectations To involve the family in care and welfare	Appreciate each other and accept diversity Do it together: talk with each other, ask for help Evaluate and share knowledge Celebrate success Listen to each other Get to know each other (better)

**Table 4 healthcare-11-01840-t004:** List of themes to work on in the design phase.

1 One team, working together, staff needs, respect 2 Communication/collaboration between residents, colleagues, other disciplines, respect 3 Heart for the residents/family, respect 4 Working with head and heart; trust, love, attention, patience, respect 5 More time for residents and family 6 Know your resident, person-centered care 7 Freedom for residents and staff 8 Freedom and choices for residents, family, and staff 9 Better on-site facilities for residents, family, and staff 10 More facilities for residents-better preconditions for staff

**Table 5 healthcare-11-01840-t005:** Ten appreciative questions for the design phase.

1. In the desired situation, what exactly does the implementation of your theme look like? 2. What is going well, and do you want to keep? 3. What is going well but should be improved? 4. What is not happening yet, but should be? 5. What do you want to say goodbye to? 6. What can be achieved in the short-term? 7. What can be achieved in the middle-term? 8. What can be achieved in the long-term? 9. What are the ideal conditions for growth? How can we achieve them? 10. What are potential obstacles to growth? And how can they be removed?

**Table 6 healthcare-11-01840-t006:** Outcomes of the destiny phase.

First AI Session	Second AI Session
1. ...Being friendly to residents and colleagues 2. ...Give and receive feedback according to the rules of feedback 3. ... Express an appreciation to each other at least once a day 4. ... Consult with each other: making work arrangements, asking if and what help is needed 5. ... Know the residents by reading (reports and agenda) and share experiences you have had with the residents with each other 6. ... Dare to learn from each other 7. ... Talking positively about residents, even with complex behavior	1. ... Respect each other: we greet each other, we listen to each other 2. ...Giving and receiving feedback according to the rules for feedback 3. ...Speak to each other first before we contact the manager 4. ...Do not judge or condemn in advance, but talk to each other, and ask each other questions

## Data Availability

The data presented in this study are available on request from the corresponding author.

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
