# Peer review of "Learning and Developing Together for Improving the Quality of Care in a Nursing Home, Is Appreciative Inquiry the Key?"

_healthcare, 2023, doi:10.3390/healthcare11131840_

Round 1

Reviewer 1 Report

The article has an important topic to address in a scientific review: the quality of nursing care and its different determinants (clients, professionals, care models), both individual and organizational. Thereby, the authors have presented a study to introduce the idea/approach that there are methodologies, namely Appreciative Inquiry, simultaneously interventive and investigative, and mixed. Methodologies that are able to translate scientific knowledge into intervention practices, and vice-versa. The qualitative approach of science also has this goal, which is difficult to deploy. Despite the results achieved, which are always unforeseen, authors have reported how those methodologies are implemented very well. Nevertheless, I would like to suggest minor changes that aim to clarify the information provided by the authors and make it easier to read for people who are not familiar.

Introduction

·       Create three subsections: 1.º Quality care in nursing homes; 2.º Appreciative Inquiry (methodology); 3.º Why Appreciative Inquiry can improve quality of care in nursing homes (reasons and evidence).

·       Review sentence punctuation: lines 45 and 46.

·       Suggestion: Authors don’t need to refer “PAR” in the introduction; instead, they should frame AI in the social theories of change, as originally proposed by David Cooperrider and Suresh Srivastva in 1987. See: https://aipractitioner.com/  

Material and Methods

·       Review the subsections. Suggestions:

o   2.1. Context, Purposes, and Methods: where the study was implemented? What are the purposes (research and social change)? Methods to collect and analyse data? PAR should be explained in this section.

o   2.2 Recruitment, Collection & Analysis, and Validation: Why that organization and not another? How methods were used? How do authors validate the data captured?

o   2.3. Limits and Ethics: Personally, I totally disagree with the ethics approach. The study should be approved by one ethical committee. The ethics approval cannot be replaced by the participants' permission. There are two different analyses. The ethical committee assesses risks and proposes mitigation measures.

o   Suggestion: The SRQR is not a methodology, but a set of criteria to report it. Thereby, it is missing the methodology approach: case study? Research-action? Ethnography?

o   Suggestion: Resume the information in a Table and a Figure.

·       Data Analysis (AI process vs Evaluation AI)

o   Clarify what is part of the AI process and what is part of the research. Data analysis is the process of evaluating data collected independently and scientifically. For instance, “content analysis” is a scientific method to analyze qualitative data adopting a technical tool or not (e.g., Nvivo or Maxqda).

Results

·       Suggestion: It will be easier if data are systemized/organized in tables and figures. Now, it’s very difficult to understand what the results are and what are the authors' explanations, but also when sections initiate.

Author Response

Dear Reviewer,

Reviewer 1

Comments and Suggestions for Authors

The article has an important topic to address in a scientific review: the quality of nursing care and its different determinants (clients, professionals, care models), both individual and organizational. Thereby, the authors have presented a study to introduce the idea/approach that there are methodologies, namely Appreciative Inquiry, simultaneously interventive and investigative, and mixed. Methodologies that are able to translate scientific knowledge into intervention practices, and vice-versa. The qualitative approach of science also has this goal, which is difficult to deploy. Despite the results achieved, which are always unforeseen, authors have reported how those methodologies are implemented very well. Nevertheless, I would like to suggest minor changes that aim to clarify the information provided by the authors and make it easier to read for people who are not familiar.

Response: Thank you so much for your positive assessment of our manuscript. We respond to your feedback points in blue text. We hope that we have been able to improve the manuscript sufficiently.

Introduction

Create three subsections: 1.º Quality care in nursing homes; 2.º Appreciative Inquiry (methodology); 3.º Why Appreciative Inquiry can improve quality of care in nursing homes (reasons and evidence).

Response: We followed your good suggestion and created three subsections in the introduction:

-Learning and quality of care in nursing homes

-Appreciative Inquiry

-The improvement of quality of care using Appreciative Inquiry

Review sentence punctuation: lines 45 and 46.

Response: Punctuation has been reviewed and adjusted.

Suggestion: Authors don’t need to refer “PAR” in the introduction; instead, they should frame AI in the social theories of change, as originally proposed by David Cooperrider and Suresh Srivastva in 1987. See: https://aipractitioner.com/                                                                                                                          

Response: PAR has been removed from the introduction, along with the reference to the study by Snoeren et al. (2016). The explanation about PAR is now described in the section Materials and Methods (paragraph 2.1 Design), including a reference to the social theories of change as proposed by Cooperrider et al.

We added the following text:

“Within a participative action research design (PAR), AI was chosen as the method for this study.  AI, like PAR, is an action research methodology which involves its subjects as co-researchers [23], is based on critical reflective practice [15] and fosters collective action and social change [24]. In distinction to conventional action-research, the knowledge-interest of AI lies not so much in problem solving but in social innovation from an appreciative approach [25]. The four phases of AI were completed in three sessions, and the five principles of AI were applied appropriately [15]. The sessions also incorporated the elements of AI group sessions as described by Masselink et al. [26]: 'whole system in the room' (i.e., everyone who has an interest in the topic participates), striving for a plurality of perspectives, empowerment for all, assuming self-direction, and finally seeking broad agreement for decision-making.”

Material and Methods

Review the subsections. Suggestions:

o   2.1. Context, Purposes, and Methods: where the study was implemented?

Response: The location was blinded for review. In this revised version the name of the organization and the location is shown.

What are the purposes (research and social change)?

Response: The purposes are described in the introduction:

“The goals of this study are: a) to investigate the impact of a complete AI cycle on learning and improving quality of care in a nursing home, and b) to explore the experiences following the entire cycle of AI with a care team in a Dutch nursing home”.

Methods to collect and analyse data?

Response: Both collection and analyse data are specified in this revised version. Two tables have been added to the data collection for concrete purposes. One for each phase of AI and one for the evaluation of the AI sessions.

2.3.      Data collection

Data was collected throughout the AI process by the researcher, the facilitators of the AI process, the participants in the team sessions, and in the workgroup of the research project. A variety of (qualitative) data collection methods have been used as appropriate to the method of AI, including interviews (e.g., discover), group discussions (e.g., discover, design, destiny), creative methods (dream), and storytelling (e.g., discover, dream). Outcomes were written on whiteboards, which have been preserved, and were worked out by the facilitators. Recordings of the explanation of the dream phase were transcribed. The data collection methods of the AI sessions are outlined in Table 1.

Table 1: Overview of AI phases and data collection

Phase

Data collection

Discover

  • Individual interview notes from the interviews in pairs (25 in the first session and 45 in the second session)
  • Seven summaries of key indicators from the interviews made in pairs by subgroups of 8 people (three in the first session and four in the second session)
  • Per session one paper with the key indicators for good care and good cooperation with the family and as a care team that apply to the attendees
  • One summary paper with the indicators of good care and good cooperation with family and in the care team from both sessions

Dream

  • Seven artworks from two sessions (three in the first session and four in the second session)
  • Seven stories about the artworks captured as images and transcripts

Design

  • Eight elaborations on paper of themes that those present have taken from the film shown, containing the seven works of art from the dream phase (four each session)
  • Two summary papers with themes taken from the film by the attendees in a session about the seven works of art of the dream phase (one each session)
  • Ten change proposals (five each session)

Destiny

  • Six notes from subgroups on what is going well in the collaboration and communication and what can be improved (three each session)
  • Six elaborations of a top three with cooperation agreements (three each session)
  • Two summary papers with cooperation agreements by the attendees in a session (one each session)
  • One list of cooperation agreements as a result of both sessions

To evaluate the AI process, the three facilitators interviewed staff members. To ensure that we got a representative sample from the participants of the AI sessions, some respondents were chosen deliberately, the psychologist, the activity coordinator, and management, for example. The interviews of the care personnel were done with employees who were working when the facilitators were on site (convenience sample). Then the planning was checked to plan interviews with the care workers who had not yet been interviewed. This was done until data saturation was reached. Care workers were interviewed individually or, for efficiency reasons, in small groups. See for specification of the interviews table 2.

Table 2: Specification of interviews to evaluate the AI sessions

Interviewer

Interviewees

Team coach

Interview with one care member

Three interviews with two care members

Interview with a care member and a facility employee

Interview with four care members

Interview with a ward assistant

Team member

Interview with the psychologist

Interview with the activity coordinator

Researcher

Six interviews with one care member

Interview with a care member and a facility employee

Researcher, team coach

Interview with management: location manager, two team leads and a quality-nurse

Researcher, team coach, team member

Interview with each other in which 2 members have interviewed the 3rd member

Semi-structured interviews were chosen to enable the participants to ask questions if

 Necessary [27]. An interview protocol was drawn up to ensure the three facilitators conducted the interviews in the same way, which benefits the quality of the data (see Supplementary Material A).

The three facilitators themselves also did an evaluation of the AI sessions by interviewing each other based on the topic list that was also presented to the team members.

All interviews were conducted orally, audiotaped and then transcribed.

This study is conducted following the Standards for Reporting Qualitative Research (SRQR) [28].

PAR should be explained in this section.

Response: PAR is moved from the introduction to design (2.1) and is elaborated in this section. We refer to a previous response to your comment (Introduction; third point).

2.2 Recruitment, Collection & Analysis, and Validation:

Why that organization and not another?

Response: This organization was chosen because the second author is affiliated with the organization as a professor of Health and well-being of frail elderly. In addition, the organisation encourages conducting scientific research in the care for older people.   

We now added this to the article as follows:

2.2.      Context and participants

The study took place from September 2021 until July 2022 in Bovenkerk, a nursing home in the Netherlands that belongs to Zonnehuisgroep Amstelland. This organization was chosen because the second author is affiliated with the organization as a professor of Health and well-being of frail elderly. Furthermore, the organisation encourages conducting scientific research in the care for older people”.

How methods were used?

Response: See above Material and Methods- 2.1: Methods to collect and analyse data.

How do authors validate the data captured?

Response: To clarify this validation, the analysis part has been rewritten, both the part of the AI sessions and the part about the evaluation of AI.  We added the paragraph ‘Rigor’. The new text is as follows:

“2.4.      Data analysis

2.4.1.     AI process

The data analysis involved a participative approach based on Appreciative Inquiry strategies as reflection, and deductive and inductive coding [29]. In each AI session and in each AI phase, the facilitators reflected orally with the participants on the data using appreciative questions such as: Is this a good representation? Is it complete? Is there anything missing? Does anyone have any reservations? Can everyone agree with this? In this way, the data was immediately member-checked [30], and was therefore validated and served as an input for the next AI phase.

In addition, the facilitators and the working group reflected on the outcomes and the process of both parallel sessions going through the data, interpreting possible meanings and identifying important themes. These themes were brought back into the AI sessions to check whether the interpretation of the facilitators and working group matched the interpretation of the team members.

2.4.2.     Evaluation AI

The steps of thematic analysis by Braun and Clarke [31] were used to analyze the data from the interviews. The researcher openly coded the interviews in Maxqda 2022 and put them in themes. The other two interviewers read the transcripts, reviewed the codes and the themes the researcher made in Maxqda. The codes and themes were discussed by the three interviewers until consensus was reached.

2.5. Rigor

Rigor was considered in different ways in the research. In the setup of the research, a specific trio of facilitators was chosen, namely the researcher, the team coach, and a team member of the location. In the process of data collection and data analysis, they were able to inspire, check, and correct each other. Secondly, an experienced external AI facilitator was coaching the three facilitators, ensuring the careful completion of the analysis steps in an AI phase and the proper follow-up of the various AI phases were guaranteed. A third step was to ask the working group to what extent the outcomes of the AI sessions were recognized. They confirmed that the outcomes were consistent with the experiences on the location. 

During the AI sessions, attention was also paid to rigor by asking the participants in all steps whether they recognized themselves in the (intermediate) outcomes (member check) (Birt el al., 2016).  Finally, the steps and considerations in data decisions have also been recorded and described for reasons of transparency and replicability of the research. To keep the article readable, those steps and decisions are described in the results.”

In the results, concreteness was added to the various steps of AI. For example, this addition was made to the following text:

In September 2021, two parallel team sessions were planned. The sessions included 25 and 45 participants, respectively. After an explanation about the 4D-AI cycle, we formed groups of two who held an appreciative interview with each other about situations where they had provided good care for the residents and their family and worked well with their coworkers. The pairs were composed by one of the facilitators, the team member, so that as many different perspectives as possible would meet in the interview, taking into account the degree of courage of team members to speak out. Each participant was given an A4 with instructions on it containing the themes to be covered and examples of appreciative questions. Instructions were also given as to what the interviewer should write down from the interview (for these instructions, see supplementary Material B).

2.3. Limits and Ethics:

Personally, I totally disagree with the ethics approach. The study should be approved by one ethical committee. The ethics approval cannot be replaced by the participants' permission. There are two different analyses. The ethical committee assesses risks and proposes mitigation measures.

Response: The study protocol was discussed and approved by the research committee of the healthcare organization (Zonnehuisgroep Amstelland). In addition, medical ethics approval was not necessary, because particular treatments or interventions were not offered or withheld from respondents. The integrity of the respondents was not encroached upon as a consequence of participating in this study, which is the main criterion in medical-ethical procedures in the Netherlands. We refer to the reference [32].

Suggestion: The SRQR is not a methodology, but a set of criteria to report it. Thereby, it is missing the methodology approach: case study? Research-action? Ethnography?                                                                  

Response: You are right. Therefore, we have moved the SRQR to the description of the data collection (2.3). The methodology approach is rewritten in the section design:

Suggestion: Resume the information in a Table and a Figure.

Response: Tables have been added in the data collection and data analysis sections (Material and Methods- 2.1: Methods to collect and analyse data)       

Data Analysis (AI process vs Evaluation AI)

Clarify what is part of the AI process and what is part of the research. Data analysis is the process of evaluating data collected independently and scientifically. For instance, “content analysis” is a scientific method to analyze qualitative data adopting a technical tool or not (e.g., Nvivo or Maxqda).

Response: See above (Material and Methods- 2.1: Methods to collect and analyse data)

Results

Suggestion: It will be easier if data are systemized/organized in tables and figures. Now, it’s very difficult to understand what the results are and what are the authors' explanations, but also when sections initiate.       

Response: We added two tables in section 2 Materials and Methods. In consideration of your comment, we have also looked into adding tables to section 3 Results, but we do not see any possibilities. If you have any suggestions, we would be happy to hear.

For more details, please see the revised manuscript.

Reviewer 2 Report

1. Overall, it seems necessary to revise the sentence concisely. If necessary, it would be good to get a review from an English expert.
2. The subject of audit inquiry is novel and seems appropriate as a positive way of strengthening the nursing quality of Nursing Home. However, there are quite a few parts to be supplemented in terms of expression and the structure of the paper.
3. Please clearly describe the research design in the research design. Following Standards for Reporting Qualitative Research is something that should be mentioned later in the data collection or analysis method.
4. Where are the saturation criteria for the data mentioned?
5. When marking AI, please check if full-term and abbreviations comply with the regulations for posting to the journal.
6. There is a lack of technology for research methods based on the content of the text on the research method. It is biased toward AI protocols, which are mixed with descriptions of qualitative research methods, confusing readers.
7. The resulting description is free to form. In qualitative research, the analysis of data should be objectified and derived in consideration of rigor.

It seems necessary to revise the sentence concisely. If necessary, it would be good to get a review from an English expert.

Author Response

Dear Reviewer,

Reviewer 2

Comments and Suggestions for Authors

Thank you so much for your useful feedback. We respond to your feedback points in blue text. We hope that we have been able to improve the manuscript sufficiently.

  1. Overall, it seems necessary to revise the sentence concisely. If necessary, it would be good to get a review from an English expert.

Response: The manuscript has been checked by a native speaker. If desired, we can provide a certificate

  1. The subject of audit inquiry is novel and seems appropriate as a positive way of strengthening the nursing quality of Nursing Home. However, there are quite a few parts to be supplemented in terms of expression and the structure of the paper.

Response: In the revised version, subheadings have been written in the introduction to organize the text. Tables have been added to the data collection and data analysis to concretize the methods used, and also to provide more structure to the text.

  1. Please clearly describe the research design in the research design.

Response: The research design has been rewritten. The new text is as follows:”

“Within a participative action research design (PAR), AI was chosen for this study.  AI, like PAR, is an action research methodology which involves its subjects as co-researchers [23], is based on critical reflective practice [15] and fosters collective action and social change [24]. In distinction to conventional action-research, the knowledge-interest of AI lies not so much in problem solving but in social innovation from an appreciative approach [25]. The four phases of AI were completed in three sessions, and the five principles of AI were applied appropriately [15]. The sessions also incorporated the elements of AI group sessions as described by Masselink et al. [26]: 'whole system in the room' (i.e., everyone who has an interest in the topic participates), striving for a plurality of perspectives, empowerment for all, assuming self-direction, and finally seeking broad agreement for decision-making”.

Following Standards for Reporting Qualitative Research is something that should be mentioned later in the data collection or analysis method.

Response: We have moved this Standard to the data collection.

  1. Where are the saturation criteria for the data mentioned?

Response: The description of the analysis now provides more information about data saturation. See the modified text below.

“2.4.      Data analysis

2.4.1.     AI process

The data analysis involved a participative approach based on Appreciative Inquiry strategies as reflection, and deductive and inductive coding [29]. In each AI session and in each AI phase, the facilitators reflected orally with the participants on the data using appreciative questions such as: Is this a good representation? Is it complete? Is there anything missing? Does anyone have any reservations? Can everyone agree with this? In this way, the data was immediately member-checked [30], and was therefore validated and served as an input for the next AI phase.

In addition, the facilitators and the working group reflected on the outcomes and the process of both parallel sessions going through the data, interpreting possible meanings and identifying important themes. These themes were brought back into the AI sessions to check whether the interpretation of the facilitators and working group matched the interpretation of the team members.

2.4.2.     Evaluation AI

The steps of thematic analysis by Braun and Clarke [31] were used to analyze the data from the interviews. The researcher openly coded the interviews in Maxqda 2022 and put them in themes. The other two interviewers read the transcripts, reviewed the codes and the themes the researcher made in Maxqda. The codes and themes were discussed by the three interviewers until consensus was reached.

2.5. Rigor

Rigor was considered in different ways in the research. In the setup of the research, a specific trio of facilitators was chosen, namely the researcher, the team coach, and a team member of the location. In the process of data collection and data analysis, they were able to inspire, check, and correct each other. Secondly, an experienced external AI facilitator was coaching the three facilitators, ensuring the careful completion of the analysis steps in an AI phase and the proper follow-up of the various AI phases were guaranteed. A third step was to ask the working group to what extent the outcomes of the AI sessions were recognized. They confirmed that the outcomes were consistent with the experiences on the location. 

During the AI sessions, attention was also paid to rigor by asking the participants in all steps whether they recognized themselves in the (intermediate) outcomes (member check) (Birt el al., 2016).  Finally, the steps and considerations in data decisions have also been recorded and described for reasons of transparency and replicability of the research. To keep the article readable, those steps and decisions are described in the results.”

In the results, concreteness was added to the various steps of AI. For example, this addition was made to the following text:

In September 2021, two parallel team sessions were planned. The sessions included 25 and 45 participants, respectively. After an explanation about the 4D-AI cycle, we formed groups of two who held an appreciative interview with each other about situations where they had provided good care for the residents and their family and worked well with their coworkers. The pairs were composed by one of the facilitators, the team member, so that as many different perspectives as possible would meet in the interview, taking into account the degree of courage of team members to speak out. Each participant was given an A4 with instructions on it containing the themes to be covered and examples of appreciative questions. Instructions were also given as to what the interviewer should write down from the interview (for these instructions, see supplementary Material B).

  1. When marking AI, please check if full-term and abbreviations comply with the regulations for posting to the journal.

Response: We will check with the editor how we can best describe Appreciative Inquiry or AI in the manuscript.

  1. There is a lack of technology for research methods based on the content of the text on the research method. It is biased toward AI protocols, which are mixed with descriptions of qualitative research methods, confusing readers.

Response: In the revised version, more distinction has been made between the phases of AI and the research methods used, for example by including a table that shows which data collection method has been used for each phase of AI. See the revised text below.

2.3.        Data collection

Data was collected throughout the AI process by the researcher, the facilitators of the AI process, the participants in the team sessions, and in the workgroup of the research project. A variety of (qualitative) data collection methods have been used as appropriate to the method of AI, including interviews (e.g., discover), group discussions (e.g., discover, design, destiny), creative methods (dream), and storytelling (e.g., discover, dream). Outcomes were written on whiteboards, which have been preserved, and were worked out by the facilitators. Recordings of the explanation of the dream phase were transcribed. The data collection methods of the AI sessions are outlined in Table 1.

Table 1: Overview of AI phases and data collection

Phase

Data collection

Discover

  • Individual interview notes from the interviews in pairs (25 in the first session and 45 in the second session)
  • Seven summaries of key indicators from the interviews made in pairs by subgroups of 8 people (three in the first session and four in the second session)
  • Per session one paper with the key indicators for good care and good cooperation with the family and as a care team that apply to the attendees
  • One summary paper with the indicators of good care and good cooperation with family and in the care team from both sessions

Dream

  • Seven artworks from two sessions (three in the first session and four in the second session)
  • Seven stories about the artworks captured as images and transcripts

Design

  • Eight elaborations on paper of themes that those present have taken from the film shown, containing the seven works of art from the dream phase (four each session)
  • Two summary papers with themes taken from the film by the attendees in a session about the seven works of art of the dream phase (one each session)
  • Ten change proposals (five each session)

Destiny

  • Six notes from subgroups on what is going well in the collaboration and communication and what can be improved (three each session)
  • Six elaborations of a top three with cooperation agreements (three each session)
  • Two summary papers with cooperation agreements by the attendees in a session (one each session)
  • One list of cooperation agreements as a result of both sessions

To evaluate the AI process, the three facilitators interviewed staff members. To ensure that we got a representative sample from the participants of the AI sessions, some respondents were chosen deliberately, the psychologist, the activity coordinator, and management, for example. The interviews of the care personnel were done with employees who were working when the facilitators were on site (convenience sample). Then the planning was checked to plan interviews with the care workers who had not yet been interviewed. This was done until data saturation was reached. Care workers were interviewed individually or, for efficiency reasons, in small groups. See for specification of the interviews table 2.

Table 2: Specification of interviews to evaluate the AI sessions

Interviewer

Interviewees

Team coach

Interview with one care member

Three interviews with two care members

Interview with a care member and a facility employee

Interview with four care members

Interview with a ward assistant

Team member

Interview with the psychologist

Interview with the activity coordinator

Researcher

Six interviews with one care member

Interview with a care member and a facility employee

Researcher, team coach

Interview with management: location manager, two team leads and a quality-nurse

Researcher, team coach, team member

Interview with each other in which 2 members have interviewed the 3rd member

Semi-structured interviews were chosen to enable the participants to ask questions if necessary [27].  An interview protocol was drawn up to ensure the three facilitators conducted the interviews in the same way, which benefits the quality of the data (see Supplementary Material A).

The three facilitators themselves also did an evaluation of the AI sessions by interviewing each other based on the topic list that was also presented to the team members.

All interviews were conducted orally, audiotaped and then transcribed.

This study is conducted following the Standards for Reporting Qualitative Research (SRQR) [28].

  1. The resulting description is free to form. In qualitative research, the analysis of data should be objectified and derived in consideration of rigor

Response: The analysis description provides more clarification on how the analysis was carried out (see the answer to your fourth point of feedback).  As mentioned in the analysis, we chose to describe the steps taken more concretely in the results with which we want to show that attention has been paid to rigor.

For example, we added the following text to the original sentence:  

Example 1:

In two sessions, seven artworks were made. Each group told the story of their artwork: What is it about? Why this artwork? The others listened and asked questions. The facilitators encouraged participants to ask questions or asked in-depth questions of their own to the presenting groups so that the meaning of the artwork was well understood.

Example 2:

In each session, two participants were asked to turn the two overviews from both sessions into one single list of agreements that would be appropriate and relevant for the entire location. The researcher supervised this process. This eventually resulted in nine collaborative agreements. These nine agreements were presented to the working group and discussed in the team meeting. Everyone in the team agreed with these nine agreements. The location had these agreements printed on posters and pocket cards, see Figure 2.

Another example of concreteness was already mentioned in the fourth feedback point.

Comments on the Quality of English Language:

It seems necessary to revise the sentence concisely. If necessary, it would be good to get a review from an English expert.       

Response: We refer to our response in your first feedback point.

Other changes made in the manuscript:

*Additional points were made in the discussion regarding rigor and impact.

The AI process has been completed in close consultation with the working group and the participants, which guarantees the participatory aspect of the research [36]. However, the triangulation of qualitative data collection methods may have provided the data with a deeper meaning and helped to minimize the limitations of each method [27]. The reliability of the findings was increased by a continuous member check in the sessions, in which the outcomes of a session were always coordinated with those present and thus validated [30]. Due to the methods used in the AI sessions, the facilitators have strived as much as possible to hear all opinions, to understand what is being said by asking many questions and, finally, to arrive at supported decisions. To check whether everyone's opinion has really been taken into account, analyzing all completed forms and whiteboards would have been another option. We therefore recommend this check for a subsequent study where AI is used.

The AI process was followed at one location with one team, which makes the generalizability of the workable principles of AI small. Therefore, it would be valuable to do follow-up research in more locations and teams to see if the AI process leads to the same experiences and outcomes.

Due to the termination of the research project after the third AI session, we were unfortunately unable to retrieve data from the evaluation that would have been done after the implementation of the agreements made about communication and collaboration. With this evaluation we could have said more about the impact of going through the AI process.

Watkins et al. [17] propose using the appraisal tool by Bushe and Kassam [37] to measure the impact of an AI process, in particular: Have assumptions changed? Has new knowledge been generated that has caused a radical shift in the way things are done? And, has a platform been found to move forward, for example by entering into and resolving conflicts? Judging by this appraisal tool, the impact of going through the AI cycle in this study is small because it is not yet possible to make firm statements about the improvement of care. However, there are proposals for change that can form the basis for further professionalization of the team and the improvement of care. It is up to the location to continue to use the AI philosophy and to work together and learn in this way for the benefit of the quality of care.

*And the conclusion slightly changed:

To provide good care to residents in a nursing home, it is necessary to continue learning—both as an individual and as a team. AI is increasingly being used as a method to effect change in healthcare. This study investigated the experiences of a nursing home’s teams while going through the entire AI cycle and asked whether this has resulted in an improvement in care. The participants experienced the AI method as an enthusing method that brought colleagues together and created connections by looking together at where people want to go and how to get there. Going through the AI cycle has resulted in a better atmosphere in the team with increased psychological safety, which is a prerequisite for learning [10]. This study has shown that AI could be a valuable way to support team learning and development. Follow-up research in Dutch nursing homes on the use of AI as a learning and development method is recommended to collect more evidence about the process and impact of AI.

For more details, please see the revised manuscript.

Reviewer 3 Report

Thank you for this very novel methodology described in the article. 

Some comments and editing:

Introduction

Line 53: the authors mention "necessary conditions" for an organization to become learning organization. There is a place to cite those conditions within the manuscript and not only citing other articles. 

Materials and method

part 2.3 lines 167-168. If the methodology is interviews, why does the authors regard participants saying most of them have difficulties reading and in ICT using. There isn't any connection between those parts and maybe it should be explained in more details. 

Also, it is mentioned that the researchers conducted 21 interviews. why 21? how were the interviewees chosen? 

The authors say that they gave special attention for interviewing in the same way (line 170). But, in line 172-173 it is mentioned that there were individual interviews as well as small group interviews. In is not the same and it should be clarified - who interviewed individually and who attended group and why. 

Results

Part 3.3 line 236 - the authors mention that they formed new groups at the same team session. why?

Part 3.4 lines 244-245 there were 2 study groups one with 25 participants and one with 45 participants. Why the size differences? it should be explained. 

Part 3/6 Again, employees that were interviewed - who chose them? why them? how they been chosen?

Author Response

Dear Reviewer,

Reviewer 3

Thank you for this very novel methodology described in the article. 

Response: Thank you so much for your positive assessment of our manuscript. We respond to your feedback in blue text. We hope that we have been able to improve the article sufficiently.

Some comments and editing:

Introduction

Line 53: the authors mention "necessary conditions" for an organization to become learning organization. There is a place to cite those conditions within the manuscript and not only citing other articles. 

Response: In the revised version we cited the conditions as follows:

“Research in learning organizations distinguish multi levels in the organization to be important. Gavin et al. [10] revealed the three building blocks of organizational learning; leadership that reinforces learning (engage in active questioning and listening, demonstrate willingness to entertain alternative viewpoints), learning processes (experimentation, information collection and transfer, education and training, analysis) and a supportive learning environment (psychological safety, openness to new ideas, time for reflection, appreciation of differences). Decuyper et al. ([11] mentioned the levels of organization (strategy, involved leadership), the team (team leadership, management) and the individual (self-efficacy, being flexible and motivated).

The above elements of learning can also be found in the self-determination theory of Ryan and Desi [12]; people's intrinsic motivation can be increased if three basic psychological needs are addressed: Autonomy, sense of competence and social connectedness. Autonomy is about choices and influence. The sense of competence is the confidence in your own ability. Social connectedness is the connection with the environment, the confidence in others, a positive and safe climate where you feel free to ask questions and you are not afraid to make mistakes”. 

Materials and method

part 2.3 lines 167-168. If the methodology is interviews, why does the authors regard participants saying most of them have difficulties reading and in ICT using. There isn't any connection between those parts and maybe it should be explained in more details. 

Response: Reading and ICT skills were included as an argument for not conducting a survey. This text has been omitted in the revised version.

Also, it is mentioned that the researchers conducted 21 interviews. why 21? how were the interviewees chosen? 

Response: We added more concreteness to this part of the data collection. The revised text now reads as follows:

To evaluate the AI process, the three facilitators interviewed staff members. To ensure that we got a representative sample from the participants of the AI sessions, some respondents were chosen deliberately, the psychologist, the activity coordinator, and management, for example. The interviews of the care personnel were done with employees who were working when the facilitators were on site (convenience sample). Then the planning was checked to plan interviews with the care workers who had not yet been interviewed. This was done until data saturation was reached. Care workers were interviewed individually or, for efficiency reasons, in small groups. See for specification of the interviews table 2.

Table 2: Specification of interviews to evaluate the AI sessions

Interviewer

Interviewees

Team coach

Interview with one care member

Three interviews with two care members

Interview with a care member and a facility employee

Interview with four care members

Interview with a ward assistant

Team member

Interview with the psychologist

Interview with the activity coordinator

Researcher

Six interviews with one care member

Interview with a care member and a facility employee

Researcher, team coach

Interview with management: location manager, two team leads and a quality-nurse

Researcher, team coach, team member

Interview with each other in which 2 members have interviewed the 3rd member

Semi-structured interviews were chosen to enable the participants to ask questions if necessary [27]. An interview protocol was drawn up to ensure the three facilitators conducted the interviews in the same way, which benefits the quality of the data (see Supplementary Material A).

The authors say that they gave special attention for interviewing in the same way (line 170). But, in line 172-173 it is mentioned that there were individual interviews as well as small group interviews. In is not the same and it should be clarified - who interviewed individually and who attended group and why. 

Response: See the previous feedback point.

Results

Part 3.3 line 236 - the authors mention that they formed new groups at the same team session. why?

Response: To explain this, we added to the manuscript: “New groups have been created based on the element of AI groups session to get as many perspectives as possible (‘striving for a plurality of perspectives’), see 2.1 Design.”

Part 3.4 lines 244-245 there were 2 study groups one with 25 participants and one with 45 participants. Why the size differences? it should be explained.

Response: We added to the manuscript (results-preparation phase 3.1): “The team sessions were scheduled, and invitations were sent out to all of the employees of the location. An employee could sign up for one of the two parallel sessions. Because team members could freely register for a session, the number of participants in a session was variable”.

Part 3/6 Again, employees that were interviewed - who chose them? why them? how they been chosen?

Response: See the explanation we gave at ‘materials and method’.

For more details, please see the revised manuscript.

Round 2

Reviewer 2 Report

Please present the full term of AI in the first sentence of the abstract.

Line spacing and paragraph shape are not uniform on page 1.

It's good that the introduction is organized by sub-title, but you don't have to present the sub-title separately.

Please complete and present table 1 according to the posting regulations.

Why don't you consider changing the order of Table 1 and Table 2? It's my personal opinion.

After the content is revised, the paper is much more faithful and I think it is good to be able to check the research process in detail.
In particular, participatory research and practice in the field are valuable challenges.
However, the review of the organization and editing you see in your paper has not been sufficient. A correction to this will need to be confirmed.

Author Response

Dear Reviewer,

Please present the full term of AI in the first sentence of the abstract.

-Response: We have now written Appreciative Inquiry in full term in the 1st sentence of the summary

Line spacing and paragraph shape are not uniform on page 1.

-Response: We have now changed this in the manuscript but we assume the editor is still checking the text for proper layout.

It's good that the introduction is organized by sub-title, but you don't have to present the sub-title separately.

-Response: We deleted the sub-titles in the introduction.

Please complete and present table 1 according to the posting regulations.

-Response: We changed the tables so they fit the regulations.

Why don't you consider changing the order of Table 1 and Table 2? It's my personal opinion.

-Response: Table 1 appears earlier than Table 2 because in the study, the output of the AI sessions comes first and then the evaluation of the AI process.

After the content is revised, the paper is much more faithful and I think it is good to be able to check the research process in detail.
In particular, participatory research and practice in the field are valuable challenges.
However, the review of the organization and editing you see in your paper has not been sufficient. A correction to this will need to be confirmed.

-Response: We are happy to respond to the feedback given but we doubt that we understand the feedback correctly. You would still like to see improvements in the design and method, do we understand that correctly? Do you have suggestions for us what we could add in these parts? Thank you very much in advance.

For more details, please see the revised manuscript.